

# TurkSentGraphExp: an inherent graph aware explainability framework from pre-trained LLM for Turkish sentiment analysis

Yasir Kilic[1] and Cagatay Neftali Tulu[2]

[1] Computer Engineering Department, Adana Alparslan Turkes Science and Technology University, Adana, Turkey
[2] Software Engineering Department, Adana Alparslan Turkes Science and Technology University, Adana, Turkey

## ABSTRACT

Sentiment classification is a widely studied problem in natural language processing (NLP) that focuses on identifying the sentiment expressed in text and categorizing it into predefined classes, such as positive, negative, or neutral. As sentiment classification solutions are increasingly integrated into real-world applications, such as analyzing customer feedback in business reviews (*e.g.*, hotel reviews) or monitoring public sentiment on social media, the importance of both their accuracy and explainability has become widely acknowledged. In the Turkish language, this problem becomes more challenging due to the complex agglutinative structure of the language. Many solutions have been proposed in the literature to solve this problem. However, it is observed that the solutions are generally based on black-box models. Therefore the explainability requirement of such artificial intelligence (AI) models has become as important as the accuracy of the model. This has further increased the importance of studies based on the explainability of the AI model's decision. Although most existing studies prefer to explain the model decision in terms of the importance of a single feature/token, this does not provide full explainability due to the complex lexical and semantic relations in the texts. To fill these gaps in the Turkish NLP literature, in this article, we propose a graph-aware explainability solution for Turkish sentiment analysis named TurkSentGraphExp. The solution provides both classification and explainability for sentiment classification of Turkish texts by considering the semantic structure of suffixes, accommodating the agglutinative nature of Turkish, and capturing complex relationships through graph representations. Unlike traditional black-box learning models, this framework leverages an inherent graph representation learning (GRL) model to introduce rational phrase-level explainability. We conduct several experiments to quantify the effectiveness of this framework. The experimental results indicate that the proposed model achieves a 10 to 40% improvement in explainability compared to state-of-the-art methods across varying sparsity levels, further highlighting its effectiveness and robustness. Moreover, the experimental results, supported by a case study, reveal that the semantic relationships arising from affixes in Turkish texts can be identified as part of the model's decision-making process, demonstrating the proposed solution's ability to effectively capture the agglutinative structure of Turkish.

Corresponding author
Cagatay Neftali Tulu,
ctulu@atu.edu.tr

# INTRODUCTION

Natural language processing (NLP) has become an integral part of modern technology, powering a wide range of applications across various domains. From machine translation (*Bahdanau, 2014*) and question answering (*McCann et al., 2018*) to text summarization (*Ramesh et al., 2016*) and chatbot development (*Csaky, 2019*), NLP enables machines to understand, interpret, and generate human language. Among these applications, sentiment analysis stands out as one of the most popular and impactful tasks. It is widely used in areas such as business intelligence (*Pang & Lee, 2008*), where customer reviews and feedback are analyzed to gauge satisfaction levels, and social media monitoring (*Pak & Paroubek, 2010*), where public opinion on products, policies, or events is tracked in real time. Additionally, sentiment analysis plays a crucial role in personalized marketing (*Vinodhini & Chandrasekaran, 2012*), financial market predictions (*Bollen & Mao, 2011*), and political campaign strategies (*Tumasjan et al., 2010*) by extracting valuable insights from text data. Its widespread adoption reflects the growing need for systems that not only process language but also interpret the emotions and intentions behind it.

Despite the advancements NLP, analyzing sentiments like Turkish language presents unique challenges due to its agglutinative nature (*Oflazer, 1994*). In Turkish, words are formed by appending multiple suffixes to a root word, often resulting in long and complex word forms. This structure allows a single word to convey meanings that would require entire phrases in other languages, complicating tasks such as tokenization, parsing, and sentiment analysis (*Yuret & Türe, 2006*). Additionally, the extensive use of suffixes significantly alters the semantic and syntactic roles of words, making it difficult for models trained on non-agglutinative languages to perform effectively (*Ersahin et al., 2019*). The variability introduced by free word order and the prevalence of compound words further amplifies the complexity (*Tohma & Kutlu, 2020*). These linguistic features demand specialized approaches in NLP, such as incorporating morphological analysis and designing models that can account for the structural relationships within words. Addressing these challenges is essential for building accurate and explainable sentiment analysis for Turkish texts.

Although many studies achieving accurate results for sentiment analysis of Turkish texts have been conducted (*Çoban, Özel & İnan, 2021*; *Eryiğit & Oflazer, 2006*; *Kurt, Kisa & Karagoz, 2019*; *Aydın, Güngör & Erkan, 2019*), almost all of these models rely on black-box models, which provide limited explainability. Specifically, their explainability approach typically focus on word-level importance, highlighting each word's contribution to model predictions through feature importance scores. While useful, this approach is limited as it fails to capture the complex interactions and semantic relationships shaped by the agglutinative structure of Turkish. Consequently, there is a growing need for advanced explainability methods that go beyond word-level explanations to provide deeper insights into phrase-level interactions and the linguistic complexities of Turkish.

To analyze these research gaps in explainable Turkish sentiment analysis domain, we list several key limitations in existing approaches. First, many methods analyze words based solely on their roots, which fails to capture the unique agglutinative structure of Turkish and its semantic richness. Second, traditional token-based tabular representations of text are insufficient for generating phrase-level explainability, as they do not fully account for the relationships between words and their suffixes. Third, most learning models are presented as black-box systems without inherent explainability, limiting their interpretability. These challenges highlight the need for methods that can transform text into structured representations, process linguistic complexities effectively, and provide explainability with high fidelity to the underlying model.

To fill the identified research gaps in explainable Turkish sentiment analysis literature, we propose a novel framework named *TurkSentGraphExp*, which directly tackles the limitations of existing approaches. First, to overcome the issue of analyzing words solely based on their roots and neglecting the agglutinative structure of Turkish, *TurkSentGraphExp* leverages a pre-trained language model to generate context-aware embedding vectors that incorporate both root and suffix-level information. Second, to address the inadequacy of token-based tabular text representations for capturing relationships between words and suffixes, the framework employs graph-based text representations, enabling a structured and relational understanding of linguistic complexities. Third, to mitigate the lack of inherent explainability in traditional black-box learning models, *TurkSentGraphExp* incorporates an explainable learning mechanism capable of providing phrase-level sentiment insights.

As shown in Fig. 1, *TurkSentGraphExp* excels at identifying sentiments derived from complex semantic relationships that existing methods fail to capture. For instance, in the sentence "Ne (Neither) yemekleri (the food) güzeldi (was good), ne (nor) de (also) hizmeti (the service)," traitional methods predict a positive sentiment by focusing on individual words like güzel ("good") and their associated weights. However, these methods neglect the overall contextual relationship between the words, resulting in an inaccurate explanation, as shown on the left side of Fig. 1. In contrast, *TurkSentGraphExp* effectively incorporates graph-based text representation to model the intricate relationships between key terms, such as ne ("neither") and its connections to yemekler ("the food") and hizmet ("the service"). This graph-based approach enables the framework to correctly predict the sentiment as negative and provide a more accurate and interpretable explanation, as depicted on the right side of Fig. 1. By capturing these semantic relationships, *TurkSentGraphExp* has capability to enhance sentiment analysis for Turkish texts.

With extensive experiments on three real-world Turkish sentiment datasets, *TurkSentGraphExp* demonstrated superior explainability performance compared to state-of-the-art methods. Leveraging graph attention networks (GAT) and GATv2 architectures, it also achieved classification accuracy improvements, outperforming the second-best architecture by 0.53 on these datasets. Furthermore *TurkSentGraphExp* provided phrase-level explanations with fidelity improvements of up to 40% under varying sparsity conditions. These results highlight its effectiveness in capturing agglutinative structures

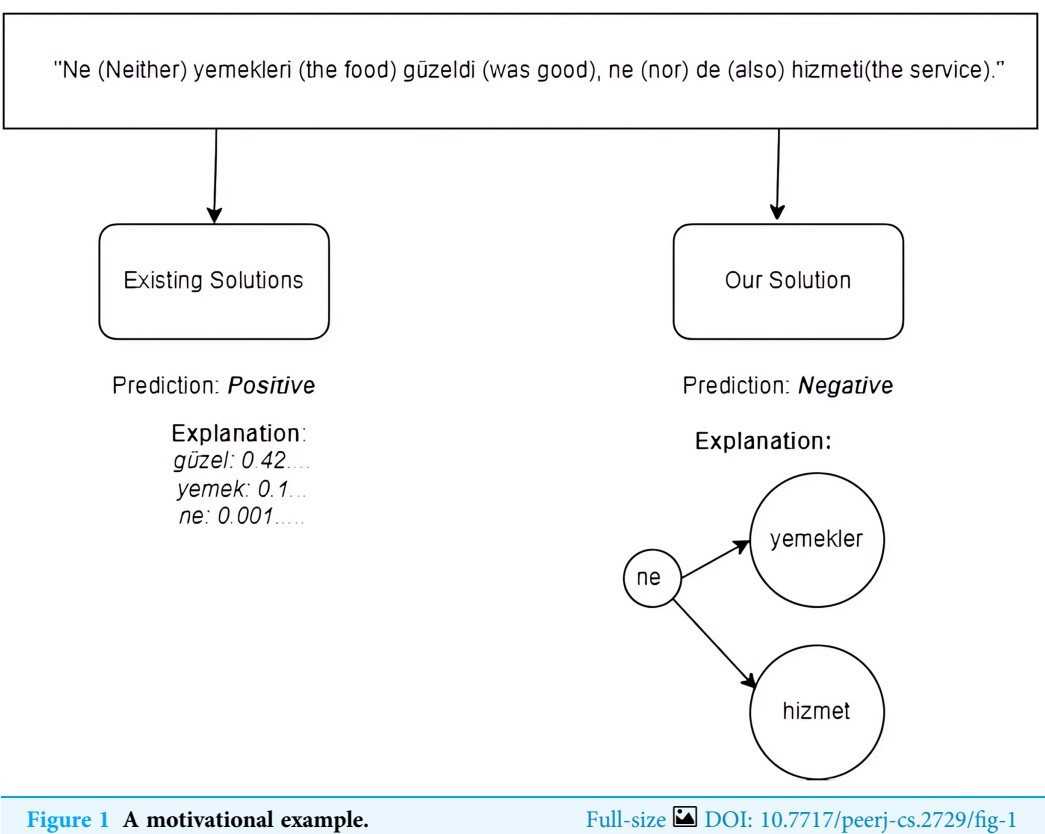

**Figure 1  A motivational example.**

and complex semantic relationships, advancing both predictive performance and explainability in Turkish sentiment analysis.

To summarize, our contributions are as follows:

- We propose *TurkSentGraphExp*, a novel framework for explainable Turkish sentiment analysis that captures both root and suffix-level information using context-aware embeddings from a pre-trained language model.
- *TurkSentGraphExp* constructs a novel graph representation using a pre-trained language model, avoiding traditional token-based methods and enabling structured, relational text representations.
- By leveraging attention based graph representation learning (GRL) architectures, *TurkSentGraphExp* inherently provides phrase-level explainability, marking the first study to achieve this for Turkish texts.
- Experiments on real-world datasets show *TurkSentGraphExp* outperforms state-of-the-art methods, achieving up to 40% higher fidelity in explainability and improving classification accuracy by margins of 0.53 over the second-best counterpart architectures.

## LITERATURE REVIEW

The agglutinative nature of the Turkish language introduces unique challenges for sentiment analysis and emotion recognition tasks in text, many of which remain

underexplored in existing studies (*Demirci, Keskin & Doğan, 2019*; *Eryiğit & Oflazer, 2006*; *Çoban, Özel & İnan, 2021*). Unlike analytic languages like English, where grammatical roles are predominantly indicated by word order and auxiliary words, Turkish relies extensively on suffixation to encode syntactic, semantic, and emotional nuances (*Oflazer, 1994*). This reliance results in the formation of long and morphologically complex words, where multiple grammatical features, such as tense, person, negation, and possession, are embedded within a single token.

Transformer-based models, such as bidirectional encoder representations from transformers (BERT) and its multilingual variants (*Devlin et al., 2018*; *Lan et al., 2020*), excel at context-aware token understanding but often struggle to capture the complex inter-morpheme and inter-word semantic relationships crucial for sentiment analysis in Turkish (*Dehkharghani, 2015*). Turkish's agglutinative structure encodes multiple layers of grammatical, semantic, and emotional information within a single word, with meanings shaped by interactions between morphemes and surrounding tokens. Sentiment understanding also depends on broader semantic relationships between words, requiring models that can represent higher-order linguistic relationships, such as dependency graphs, to effectively address these challenges.

In the next parts of this section, we review the relevant literature by categorizing the studies into two groups: Explainability Approaches for Sentiment Classification in the Turkish NLP Domain and Graph-based Explainability Approaches in the NLP Domain. The first category focuses on methods specifically developed for explaining sentiment classification tasks in Turkish, addressing the unique linguistic challenges posed by its agglutinative structure. The second category reviews graph-based approaches for explainability in NLP more broadly, emphasizing techniques that leverage graph representations to capture complex relationships between words and phrases, which are essential for understanding semantic interactions in text.

## Explainability approaches for sentiment classification task in Turkish NLP domain

In the field of natural language processing in Turkish, various studies have been conducted regarding the classification of sentiments expressed in texts (*Koksal & Ozgur, 2021*; *Ozcelik et al., 2021*; *Yıldırım et al., 2015*; *Dehkharghani, 2015*). When the presented solutions are examined, it is observed that they roughly fall into three distinct categories. The first category involves studies that consider the structure and features of the language, perform syntactic and morphological analyses, and synthesize word stems extracted with the help of a sentiment lexicon or a lexical semantic network (WordNet; *Ehsani, Solak & Yildiz, 2018*) for emotion analysis. The second category comprises machine learning and deep learning-based models that rely on labeled data and are trained on a large number of manually labeled Turkish texts. Solutions created by fine-tuning pre-trained language models with a small amount of labeled data can also be included in this category. Finally, we can list solutions that represent texts on a graph network, trained using GRL where related words are interconnected, as solutions dependent on labeled data.

In studies belonging to the first category, the distinctiveness of the Turkish language both morphologically and syntactically from other languages is highlighted, and the importance of suffixes to the semantic meaning of the words is emphasized. *Yıldırım et al. (2015)* have studied the impact of preprocessing steps (tokenization, normalization, morphological analysis, and disambiguation) on classification accuracy for Turkish texts. The study was conducted on a Turkish social media dataset, and it was observed that normalization and morphological analysis in preprocessing increased sentiment accuracy by more than 5%. Support vector machines (SVM) was used as the classification model in this study. On the other hand, *Eryiğit & Oflazer (2006)* conducted a sentiment analysis study for Turkish texts using both dictionary and machine learning approaches (naive Bayes, support vector machines, and J48). Despite the complex morphological structure of Turkish, the use of a dictionary improved classification accuracy by 7%. Furthermore, in the studies *Dehkharghani (2015)* and *Ozcelik et al. (2021)*, they developed Turkish sentiment lexicons for use in sentiment analysis of Turkish texts. In the SentiTurkNet by *Dehkharghani (2015)*, a sentiment lexicon was created by manually annotating Turkish words found in the BalkaNet (*Bilgin, Cetinoglu & Oflazer, 2004*) the Turkish WordNet as positive, negative, or neutral. This lexicon allowed sentiment analysis at the text level by combining sentiment labels from words in Turkish texts using the lexicon. Due to the limited number of conceptual expressions in SentiTurkNet and the abundance of out-of-lexicon words, methods using this lexicon gained lower performance. A more comprehensive sentiment lexicon network called HisNet (*Ozcelik et al., 2021*) was developed by labeling the sentiment polarities of 76,000 words from another Turkish WordNet called KeNet (*Ehsani, Solak & Yildiz, 2018*). While dictionary-based methods have the advantage of being more expressive compared to machine learning-based methods, the building and maintenance of a sentiment lexicon, the inclusion of new concepts, and the exclusion of out-of-lexicon words have led to lower preference for them compared to modern sentiment analysis approaches.

In the study by *Guven (2023)*, Turkish sentiment analysis was rigorously evaluated using a novel text-filtering method in conjunction with pre-trained language models. This innovative text filtering approach involves the exclusion of words within labeled texts that exhibit sentiments contrary to the assigned label. Pre-trained language models such as BERT (*Devlin et al., 2018*), A Lite BERT (ALBERT) (*Lan et al., 2020*), Efficiently Learning an Encoder that Classifies Token Replacements Accurately (ELECTRA) (*Clark et al., 2020*), and DistilBERT (*Sanh et al., 2020*) were fine-tuned and assessed on Turkish hotel and movie datasets. Notably, it was observed that when combined with the ELECTRA language model, the innovative text filtering method achieved state-of-the-art (SOTA) performance in Turkish sentiment analysis. On the other hand, *Kilic & Buyukeke (2021)* conducted a study on sentiment analysis of Turkish texts using an expressive approach known as Text-graph convolutional networks (GCN) (*Yao, Mao & Luo, 2019*), which is based on graph representation learning. To the best of our knowledge, there have been no prior instances in the literature of employing GRL for Turkish natural language processing tasks, especially in the context of sentiment analysis.

As it is evident, sentiment analysis studies conducted in Turkish language so far have predominantly relied on approaches that use word representations obtained independently from the context, disregarding the impact of compound words, phrases, affixes, and conjunctions on meaning. These approaches have been dependent on either a sentiment lexicon, a pre-trained language model, or an excessive amount of training data. In this research, we introduce a sentiment analysis framework tailored for the Turkish language. Our approach is rooted in natural language understanding and explainability. It takes into account context-sensitive words, phrases, and even linguistic affixes, considering the agglutinative nature of Turkish as a language. As a result, this framework allows for a profoundly interpretable sentiment analysis of Turkish texts.

## Graph-based explainability approaches in NLP domain

Explainability in AI (XAI) powered systems and methods is crucial for transparency, fairness, trustworthiness, and improving overall reliability and usability. It plays a pivotal role in addressing ethical, regulatory, and practical challenges associated with the deployment of deep learning (DL) models. XAI employs specific techniques and methods to enable the traceability and explainability of each decision made during the machine learning (ML) process. On the other hand, AI often reaches a conclusion using a ML algorithm, but the developers of these systems cannot fully explain how the algorithm arrived at this conclusion. This complicates accuracy, limits transparency, and hinders accountability. XAI techniques consist of three main methods: consistency in predictions, traceability, and comprehensibility of the decisions made.

In particular, explainability presents a challenge in methods used for solving NLP problems. A three-step approach for XAI in NLP models is presented (*Mishra, 2022*). These steps involve word embeddings (input level), the inner workings of NLP models (processing level), and the models' decisions (output level). One of the initial tasks at the word embedding level is to transform and visualize large-dimensional representation vectors into two or three dimensions. In this way, it becomes possible to observe how semantically related words cluster together. From this perspective, visualization at the representation level is seen as an effective XAI technique. An online approach to deriving an interpretable word embedding model (OIWE) (*Luo et al., 2020*) is one of the techniques developed in this direction. Furthermore, some approaches use external resources as XAI techniques at the embedding level. In these methods, representation vectors are created through a dictionary or a semantic lexical network, allowing semantically related words to cluster together (*Faruqui et al., 2015*; *Orhan & Tulu, 2021*).

At the processing level of XAI, there are two approaches: *Post hoc* interpretation and inherent interpretation. In the first approach, *post hoc* interpretation, efforts are made to dissect the hidden information in deep neural networks trained to solve an NLP problem syntactically and semantically. This allows us to observe the contributions of each input to both meaning and syntax. Techniques such as heatmap and t-SNE visualization can be used to visualize different semantic relationships between words in this context. On the other hand, inherently interpretable deep neural networks, such as recurrent neural networks (RNN) and long short term memories (LSTM), are trained in an explainable

manner by adding transparency constraints and discovering tree and graph-like structures. Among the highly explainable inherently interpretable methods are the Sequential Iterative Soft-Thresholding Algorithm and deep unfolding (*Hershey, Roux & Weninger, 2014*; *Wisdom et al., 2016*) methods. In the case of the new-generation LLMs, transformers are used, and various studies have been conducted to understand the internal dynamics of transformers and visualize attention weights (*Lee, Shin & Kim, 2017*; *Liu et al., 2018*). ExBERT (*Hoover, Strobelt & Gehrmann, 2019*) is a visual tool designed specifically for the explainability of the BERT model. It creates explainability in the form of attention, compilation, and summary of text representations learned depending on the content by providing masking and proximity analysis. In the study by *Wang et al. (2020)*, which focused on the GPT-2 model, a transformer-based large language model, they examined the relationship between syntactic dependency and attention weights and investigated tokens, Part of Speech (POS) tags, and head counts. They found that attention heads in the early layers focused on determiners, while in deeper layers, they focused on nouns. In the middle layers, attention was concentrated on dependency relations. As a result, they determined that heads in the early layers were more focused on position than content, while attention heads in the deeper layers were more focused on specific structures. Therefore, they concluded that long-distance relationships were captured through heads in the deeper layers. The general limitation of this approach is that attention does not have a unified definition of explainability. At the prediction level of models, approaches involving both *post hoc* and inherently interpretive methods have been adopted to enhance explainability. *Post hoc* interpretation approaches analyze the decisions made by a pre-trained model for a given input text. If the model's architecture is not transparent, this approach is known as model-agnostic, while for models with transparent architectures, it is referred to as model-specific. In the study by *Ribeiro, Singh & Guestrin (2016)*, a method called Local Interpretable Model-Agnostic Explanations (LIME) was proposed, which can be applied to the prediction layer of any machine learning model. Additionally, for NLP models at the prediction layer, XAI methods, such as perturbation and layer propagation, are also used (*Samek et al., 2019*). These methods are computationally costly since they do not require manual expert evaluation.

In the field of XAI, ongoing research explores various approaches, though standardized methods remain underdeveloped in ML, DL, and transfer learning. GRL has fewer XAI methods due to its novelty. *Yuan et al. (2022)* categorized graph neural networks (GNN) based XAI methods into two groups: instance-level (*Funke, Khosla & Anand, 2020*; *Luo et al., 2020*; *Schlichtkrull, De Cao & Titov, 2020*; *Ying et al., 2019*), which explain individual examples, and model-level (*Yuan et al., 2020*), which provide high-level insights. Their study applied these methods to sentiment analysis, constructing dependency-parsing-based graphs with words as nodes and relationships as edges. Node representations were initialized using BERT-generated vectors. GCN, graph attention networks (GAT), and graph isomorphism network (GIN) models were used for predictions, while words emphasizing sentiment were visualized for explainability. To evaluate explainability, the study used fidelity (masking key features to measure impact on accuracy), sparsity (highlighting influential features), stability (measuring accuracy consistency under small

changes), and accuracy (comparing results with synthetic data). These metrics assessed the effectiveness of graph-based XAI methods for sentiment analysis in text data.

# PROBLEM STATEMENT

In this section, we describe an explainable sentiment analysis problem in the Turkish domain. As reviewed in "Explainability Approaches for Sentiment Classification Task in Turkish NLP Domain", there are numerous studies focus on sentiment analysis for Turkish texts. These studies predominantly offer explainability based on individual word contribution or importance. However, as illustrated in the motivational example in Fig. 1, such explanation models may fail to accurately analyze the sentiment of certain texts. Furthermore, accounting for the agglutinative structure of the Turkish language is critically important for effectively identifying the sentiment of a text.

In this study, we aim to effectively address these challenges by formulating the problem as a graph representation-based learning model, overcoming the limitations of single-word importance-based explainability.

Formally, each review is represented as a graph denoted as $G(V, E)$, where $V$ represents the set of nodes and $E$ represents the set of edges. Here, $V$ consists of individual words or tokens within the review, while $E$ comprises the connections between these words based on their semantic and syntactic relationships. Each review is associated with a sentiment class label denoted as $c \in C$, indicating whether the sentiment expressed in the review is positive, negative, or neutral. To achieve explainability, related subgraphs from $G$ that provide insights into the sentiment classification process. These subgraphs highlight key features and relationships within the review that contribute to the sentiment prediction.

The task is to assign a sentiment label to each of the $n$ texts from $c$ possible classes, using an inductive approach based on graph classification. To do this, we can use Eq. (1):

$$G_i \in \{G_1, G_2, \ldots, G_n\} \tag{1}$$

where $i = 1, 2, \ldots, n$. Each $G_i$ has a sentiment label $y_i$, where $y_i \in \{1, 2, \ldots, c\}$. The goal is to learn a function $f(G_i) \rightarrow y_i$ that maps the graph representation $G_i$ to its sentiment class $y_i$. This can be done by using a graph-based ML model which can extract both the contextual and structural information from the graphs to perform sentiment classification in an inductive way. In short, the inductive approach for the task consists of creating a graph for each text and learning a function to classify these graphs into $c$ sentiment classes, enabling sentiment analysis on new texts by using graph-based models.

To ensure the model's decisions are interpretable, it is necessary to infer the importance of edges $e \in E(G_i)$ in the graph $G_i$, enabling explanations at the phrase level. Let $E(G_i)$ denote the set of edges in the graph representation $G_i$. Each edge $e \in E(G_i)$ is assigned an importance score $w(e)$, which is learned during the model training process. This allows the model to highlight the most critical edges that contribute to the sentiment classification decision. The refined goal is to learn a function in Eq. (2):

$$f(G_i, w(E(G_i))) \rightarrow y_i \tag{2}$$

where $G_i$ is the graph representation of a text, $w(E(G_i))$ represents the importance scores

of the edges in $G_i$, and $y_i \in \{1, 2, \ldots, c\}$ is the sentiment label assigned to $G_i$. This approach enhances explainability by identifying and leveraging the most significant phrases (represented by connected nodes and their associated edges) in the sentiment classification process.

## PROPOSED APPROACH

In this section, we introduce *TurkSentGraphExp* as a solution explainable sentiment analysis problem of Turkish texts described in "Problem Statement". This solution aims to handle the agglutinative structure of the Turkish language by creating graph representations through a pretrained language model, thereby establishing an inherent learning model that provides phrase-level explanations to end users.

Figure 2 illustrates the complete end-to-end workflow of the proposed model. First, the raw Turkish texts are transformed into graph representations using a pre-trained language model, as illustrated in "Graph Construction Module" and depicted in the "Graph Representation" stage of the figure. Next, as outlined in "Attentive Classification Module", these graph representations are used to train an attention-based GNN learning model under inductive settings, as shown in the "Training GNN under Inductive Settings" step in the figure. Finally, the trained GNN model is applied to predict the test data, enabling model diagnostics. Using the inherently computed attention scores, phrase-level explainability is generated, as described in "Explainability Module" and visualized in the "Attention Weights" and "Prediction" stages of the figure.

### Graph construction module

Instead of representing review documents in the typical sequential manner, we model them in graph form. There are two primary reasons for this preference. Firstly, in the case of sequential dependencies between words in comments, long-range semantic relationships may be neglected. Using such graph representation, such long-range semantic relationships are preserved. Secondly, the preference arises from the realism of explanation being based on patterns of relationships between words rather than solely on the importance of individual words.

Constructing a Turkish text graph representation involves navigating several intricate challenges, particularly concerning the generation of context-aware token embeddings, handling of suffixes, and modeling dependency edges. Firstly, creating context-aware token embeddings demands a nuanced understanding of Turkish morphology and syntax to capture the rich contextual information embedded within the language. Given the agglutinative nature of Turkish, where suffixes can significantly alter the meaning of a word, accurately representing token embeddings requires careful consideration of the surrounding context. Additionally, effectively handling suffixes poses a considerable challenge, as they play a crucial role in Turkish word formation and can vary in form and function depending on the context. Managing these suffixes while maintaining the integrity of token representations is essential for constructing an accurate graph representation. Furthermore, modeling dependency edges to capture the relationships

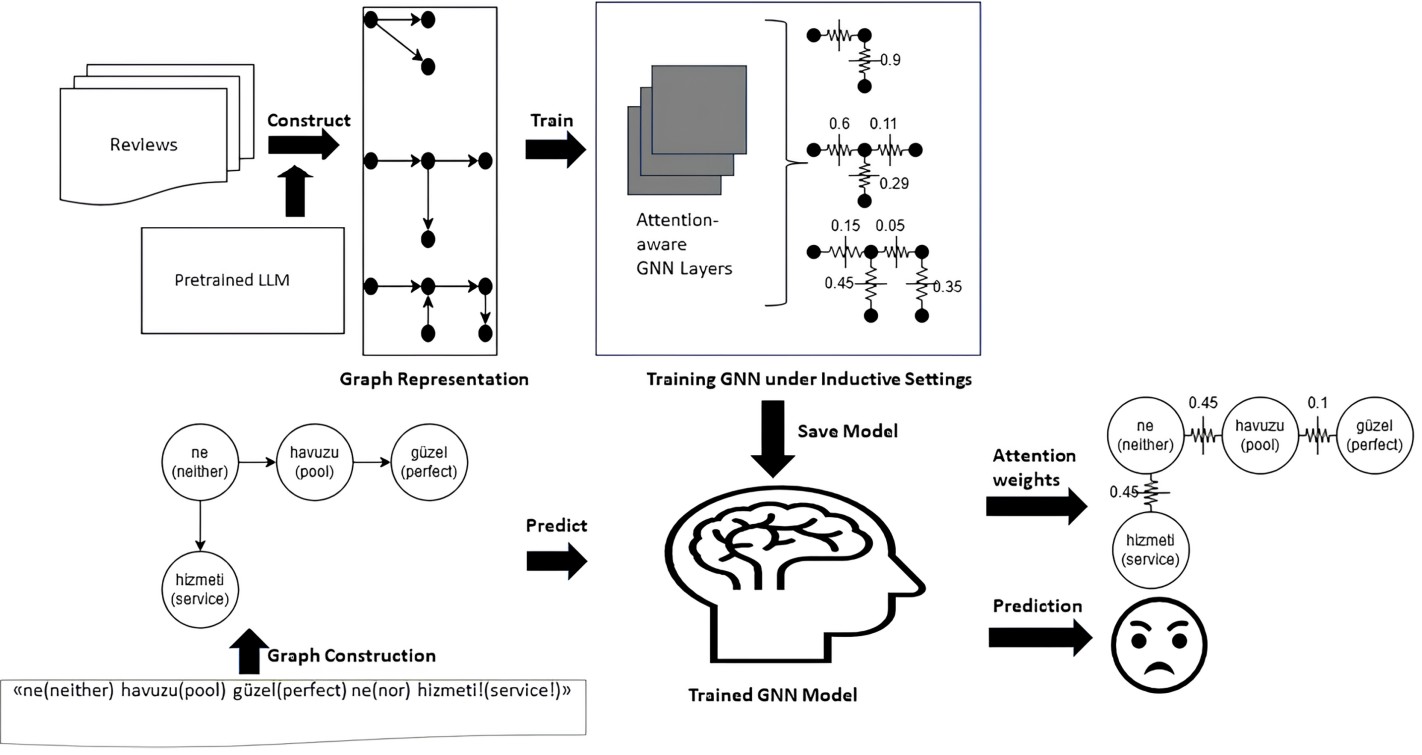

**Figure 2 Graphical workflow of the *TurkSentGraphExp*.**

between words in Turkish text requires addressing the complexities of Turkish syntax, including flexible word order and complex syntactic structures.

The construction of a text representation graph requires addressing several challenges, including generating context-aware token embeddings, handling suffix variations, and modeling dependency edges in Turkish text. Leveraging a pre-trained language model provides an effective solution to these challenges due to its robust capability to capture rich contextual information and manage the complex morphological structure of Turkish. Pre-trained models are specifically advantageous because they are trained on vast amounts of diverse text data, enabling them to generalize effectively and encode linguistic nuances such as suffix-driven semantic changes and intricate syntactic dependencies. This makes them particularly well-suited for representing the agglutinative nature and linguistic complexity of Turkish text in graph-based formats.

Moreover, the bidirectional nature and fine-tuning capability of transformer-based language models enhance their ability to capture complex linguistic relationships, such as intricate semantic interactions and suffix-driven variations in Turkish text. This enables the effective representation of Turkish text in graph structures, preserving long-range dependencies and uncovering nuanced word relationships essential for accurate and explainable sentiment analysis (*Dehkharghani, 2015*; *Htut et al., 2019*; *Limisiewicz, Rosa & Mareček, 2020*).

The inference of context-aware word embeddings using a transformer-based pre-trained language model is outlined in "Context-Aware Word Embeddings". The use of these vector values to address suffixes is described in "Handling of suffixes", while "Edge Construction" explains the process of defining inter-word relationships and determining their weights.

### Context-aware word embeddings

The calculation of context-aware word embeddings from a pre-trained BERTurk model involves several key steps. Technically, the pre-trained BERTurk model processes text sequences by first tokenizing the input text into subword units using a tokenizer function Tokenize (input), which converts the text into a sequence of tokens (CLS, $t_1, t_2, \ldots, t_n$). These tokens are then embedded into vector representations using an embedding function; Embedding ($t_i$); resulting in token embeddings Embedding (CLS), Embedding ($t_1$), ..., Embedding ($t_n$).

To compute the output vectors, BERT employs a self-attention mechanism through multi-head attention layers. Given the token embeddings, BERT constructs attention matrices Attention($t_1$), ..., Attention($t_n$) for each token. The attention matrix Attention($t_i$) for token $t_i$ measures the attention values between $t_i$ and all other tokens in the sequence.

The calculation of attention values involves computing compatibility scores between query representations $q_i$ of the current token $t_i$ and key representations $k_j$ of all other tokens $t_j$. This is achieved through the dot product softmax($q_i \cdot k_j^{\mathrm{T}}$), where softmax normalizes the scores to obtain attention weights. The resulting attention weights are used to compute weighted sums of the token embeddings, yielding the final output vectors [Output(CLS), Output($t_1$), ..., Output($t_n$)].

In summary, the model processes text sequences by tokenization, embedding, and attention mechanisms, utilizing self-attention to capture contextual dependencies and generate output vectors that encode semantic information in the text.

To adapt the inference mechanism for Turkish texts, given its agglutinative nature where word meanings rely heavily on stems and suffixes, we modify the attention mechanism. Specifically, we suggest that the attention matrix should focus solely on the left context, reflecting the structure of Turkish. This means that for the $i$-th word, only the relation with words from 1 to $i$ should be considered. This can be achieved by zeroing out the lower triangular part of the attention matrix, $i.e.$, setting $A_{ij} = 0$ if $j > i$. Consequently, the model learns from the left context while disregarding the right words.

To address the another challenge posed by the agglutinative nature of Turkish, where suffixes are inherently related to their root words, the attention mechanism is adapted to eliminate redundant attention weight learning between roots and their suffixes. Specifically, an attention masking mechanism is implemented to ensure the model focuses on meaningful inter-word relationships. During inference, this mechanism applies an attention mask, denoted as $M \in {0, 1}^{n \times n}$, where $M_{ij} = 0$ if the $j$-th word is a suffix of the $i$-th word, and $M_{ij} = 1$ otherwise. The attention matrix $A$ is then updated element-wise as $A = A \odot M$, effectively ignoring attention between root words and their corresponding

suffixes. By focusing the attention mechanism solely on root words, the framework efficiently capture meaningful inter-world relationships for graph representational purposes.

### Handling of suffixes

The handling of suffixes in the Turkish language plays a crucial role in conveying semantic and grammatical information. Stems carry core semantic meanings, while suffixes indicate grammatical features like number, case, and tense. In our research domain, especially for explainability tasks, representing words as a single vector that encapsulates both stem and suffix meanings can enhance the model's ability to capture semantic relations.

To achieve this, we propose using an *AGGREGATE* function that combines the vectors of word stems and suffixes into a unified representation. This *AGGREGATE* function can take various forms, such as sum, average, or concatenation, depending on the desired word representation strategy. Formally, let $s$ be the vector of the word stem, $e_1, e_2, \ldots, e_k$ be the vectors of the word suffixes, and $w$ be the resulting vector of the whole word. The formulation of this aggregation process is given in Eq. (3)

$$w = \text{AGGREGATE}(s, e_1, e_2, \ldots, e_k). \tag{3}$$

Let us analyze the sentiment of the sentence "Bu filmden hoşlanmadım!" (I didn't like this movie!) in a sentiment analysis context. This sentence is straightforward and expresses a negative sentiment towards the movie. The presence of the negation suffix "-madım" (didn't) clearly indicates the negation of liking, contributing to the overall negative sentiment conveyed. This example showcases how negation suffixes play a crucial role in sentiment analysis by altering the sentiment polarity of a sentence.

### Edge construction

Various graph/edge construction approaches have been proposed in the literature (*Wu et al., 2021*; *Hu et al., 2021*; *Franciscus, Ren & Stantic, 2019*). Among these solutions, the most realistic solution for representing texts in a graph format is the dependency graph (*Barbero & Lombardo, 1995*). Since it represents the syntactic dependency of words in a text document, it is a more realistic approach than heuristic methods in order to capture the meaning of the whole text document. This approach is employed to capture the syntactic relationships between words and phrases within texts. The graph creates a graphical structure where each single word in the text is represented as a node, and the dependencies between these nodes are depicted using arrows as edges. This approach allows us to better understand the relationships between words and sentence structure in natural language. By enabling a more profound and meaningful analysis of texts, the effectiveness of dependency graphs is demonstrated on NLP tasks (*Durandin & Malafeev, 2019*; *Hu et al., 2021*). Consequently, for representing texts in a graph format, the Dependency graph method stands as a suitable solution, preserving syntactic structures and aiding in a better comprehension of text semantics.

Please notice that directly constructing a dependency graph from a given text is a challenging task. This task is especially challenging for Turkish documents because

Turkish is an agglutinative language with complex morphology and word order. To construct a dependency graph from Turkish text, we need an extra model that can handle the morphological and syntactic variations of Turkish words.

Using a pre-trained language model like BERT simplifies handling complexity and enhances semantic quality compared to directly constructing dependency graphs. BERT's extensive training captures nuanced semantics and linguistic patterns, reducing the need for manual feature engineering and ensuring accurate representation of syntax, word order, and contextual relationships. Leveraging pre-trained models streamlines natural language processing tasks, particularly in complex language scenarios such as Turkish, where constructing dependency graphs requires addressing morphological and syntactic variations effectively.

In this study, to perform dependency graph edges, we propose using of a pre-trained BERT-based language model. We formulize the construction process of dependency graph as shown in Algorithm 1.

The pseudocode outlines a method for constructing a word dependency graph using a Pre-trained BERT model and a review document. It begins by initializing the necessary input parameters, including the Pre-trained model, the review document, and the number of top attended words to consider. The algorithm then initializes an empty set for vertices and edges of the graph. It iterates through each token in the attention matrix derived from the Pre-trained BERTurk model, finding the top-$k$ tokens with the highest attention values to each token. For tokens with a sufficient number of top attended words, it adds vertices to represent the tokens and edges to represent the dependencies between tokens. The resulting word dependency graph captures the semantic relationships between words based on their attention patterns in the review document, providing a structured representation of textual dependencies.

## Attentive classification module

In this module, a sentiment classification model for Turkish reviews is constructed based on the generated representation graphs.

This classification model is built using the GAT (*Veličković et al., 2017*; *Brody, Alon & Yahav, 2021*) architecture, which is a attention-based GRL model. The primary reason for this choice is the interpretability of the model's decisions based on inherently learned attention values. In other words, it can be directly interpreted without constructing any *ad-hoc* model.

GAT uses attention to focus on informative neighboring nodes when updating a node's embedding. However, a key limitation of GAT is its "static attention": the importance of neighbors is determined solely by their features, independent of the specific node we are considering. GATv2 addresses this by modifying the computation order. This allows each node to attend to its neighbors dynamically, leading to a more expressive model capable of capturing complex graph structures compared to the original GAT.

From a technical standpoint, the solution to using this model needs to be formulated as transductive or inductive. The rationale behind this choice and the notational formulation are detailed in "Formulation of Problem: Inductive *vs* Transductive". Then, in "Attention-

**Algorithm 1** Proposed graph construction through a pre-trained LLM.

1: **Input:** $M_\theta$: Pre-trained language model, $R$: Review document, $k$: Number of highest attended words to consider

2: **Output:** Word Dependency graph $G = (V, E)$

3: **Initialize:** Retrieve attention matrix $A$ from $M_{\text{BERT}}$ for review document $R$

4: Initialize a set $V$ for review words and their embeddings

5: Initialize empty set $E$ for edges

6: **for** each token $i$ in attention matrix $A$ **do**

7:     Find the top-$k$ tokens list $J_i$ with the highest attention values to token $i$

8:     **if** $J_i$ is not empty and $|J_i| \geq k$ **then**

9:         **for** each token $j$ in $J_i$ **do**

10:             Add edge $(v_j, v_i)$ to $E$

11:         **end for**

12:     **end if**

13: **end for**

14: **return** $G = (V, E)$

based GNN Design for Sentiment Classification", the construction of GAT model based on this formulation is introduced.

### Formulation of problem: inductive vs transductive

In the inductive setting, the GNNs learn to generalize from a training graph to unseen nodes or graphs in the test phase. This means that the GNNs have to capture the underlying patterns and rules that govern the graph structure and node labels. In the transductive setting, the GNNs have access to the features of all nodes in the graph, including the test nodes, but not their labels. This means that the GNNs can exploit the local neighborhood information of the test nodes to make predictions, without learning to generalize to new nodes or graphs.

A fundamental advantage of the inductive setting is that it allows GNNs to handle dynamic graphs, where new nodes or edges can be added over time. This is useful for applications where the graph structure is not fixed or known in advance.

Note that the inductive setting fits our task of sentiment classification, described in "Problem Statement", perfectly. Therefore, the proposed model is built under the inductive setting.

Formally, the task involves classifying each of the $n$ texts, composed of $c$ sentiment classes, using a graph-based inductive formulation for graph classification.

Let $G_i$ represent the graph representation of the $i$-th text document, where $i$ ranges from 1 to $n$.

Each graph $G_i$ consists of nodes and edges, capturing the structural and semantic information within the text as detailed in "Layer-Wise Embedding Evaluation".

We aim to classify each $G_i$ into one of the sentiment classes $C$. Mathematically, we can represent this task in Eq. (4):

$$G_i \in \{G_1, G_2, \ldots, G_n\} \rightarrow y_i \tag{4}$$

where $i = 1, 2, \ldots, n$. Each $G_i$ is associated with a sentiment label $y_i$, where $y_i \in C$.

The objective is to learn a function $f(G_i) \rightarrow y_i$ that maps the graph representation $G_i$ to its corresponding sentiment class $y_i$. This is achieved through a GAT-based learning model as detailed in "Attention-based GNN Design for Sentiment Classification", which can capture both the structural and contextual information encoded in the graphs to perform sentiment classification inductively.

### Attention-based GNN design for sentiment classification

After formulating the learning approach *via* graph construction, the edge-level attention-based GAT model is designed as presented in Algorithm 2.

$$e_{ij} = a^T \cdot \text{LeakyReLU}(W \cdot [h_i^l || h_j^l]) \tag{5}$$

$$\alpha_{ij} = \frac{\exp(e_{ij})}{\sum_{k \in \mathcal{N}(i)} \exp(e_{ik})} \tag{6}$$

$$h_i' = \sigma\left(\sum_{j \in \mathcal{N}(i)} \alpha_{ij} \cdot h_j\right) \tag{7}$$

$$h_{\mathcal{G}} = \text{Pooling}(h_1', \ldots, h_{\mathcal{V}}'). \tag{8}$$

Algorithm 2 presents an inductive model training approach using the GAT architecture for sentiment analysis. The algorithm takes as input a dataset $\mathcal{D}$ containing text graphs $G_i$ and their corresponding sentiment labels $Y_i$, where $i$ ranges from 1 to $N$. The goal is to train the model parameters $\theta$ to optimize sentiment classification.

To begin training, the algorithm initializes the model parameters $\theta$, including the learning rate $\eta$, optimizer $\mathcal{O}$, and loss function $f^{loss}$. Additionally, the number of epochs $T$ and batch size $B$ are set.

The dataset $\mathcal{D}$ is then split into training ($\mathcal{D}train$) and validation ($\mathcal{D}val$) sets. The algorithm iterates through epochs, where each epoch involves shuffling the training data and processing it in batches of size $B$.

In each batch iteration, the algorithm employs an advanced attention mechanism process the graphs or documents. This mechanism enhances the model's ability to capture intricate relationships and dependencies within the data. Initially, the algorithm initializes the word embeddings $H$ and gathers edge information $E$. The attention coefficients $e_{ij}$, crucial for identifying relevant features, are computed using a refined formulation inspired by GATv2's attention mechanism, as illustrated in Eq. (5). Notably, GATv2 incorporates learned attention weights $a$ to fine-tune the attention coefficients, thereby improving the model's discriminative power. Subsequently, these coefficients are transformed into attention weights $\alpha_{ij}$ through a softmax operation based on Eq. (6). This transformation ensures that the most salient features contribute significantly to the overall graph representation. The updated embeddings $H_i'$ are then derived using GATv2's advanced

---

**Algorithm 2** Inductive model training using GAT architecture.

1: **Input:** $\mathscr{D} = (G_1, Y_1), (G_2, Y_2), \ldots, (G_N, Y_N)$: Text Graphs and Sentiment Labels

2: **Output:** $\theta$: Trained Model Parameters

3: Initialize $\theta$ (*e.g.*, learning rate $\eta$), optimizer $\mathscr{O}$, loss function $f^{loss}$

4: Set epochs $T$, batch size $B$

5: Split $\mathscr{D}$ into $\mathscr{D}train$ and $\mathscr{D}val$

6: **for** $t \leftarrow 1$ to $T$ **do**                                                          ▷ Epochs

7:          Shuffle $\mathscr{D}train$

8:          **for** $i \leftarrow 1$ to $\lceil |\mathscr{D}train|/B \rceil$ **do**                  ▷ Batches

9:                  $batch \leftarrow$ Next batch of size $B$ from $\mathscr{D}train$

10:                  **for** $(G_i, Y_i) \in batch$ **do**                          ▷ Graphs/documents

11:                          $H \leftarrow G_i.V$                          ▷ Initial word embeddings

12:                          $E \leftarrow G_i.E$                          ▷ Edges for word dependencies

13:                          Compute attention coefficients $e$ using Eq. (5)

14:                          Compute attention weights $\alpha$ using Eq. (6)

15:                          Compute updated embeddings $H_i\prime$ using Eq. (7)

16:                          Pooling on $H_i\prime$ for graph-level representation (See Eq. (8))

17:                          $z_i \leftarrow \text{FC}(H_i')$, $z_i \leftarrow \text{ReLU}(z_i)$, $y_i \leftarrow \text{Softmax}(z_i)$

18:                          Compute loss $L_i = f^{loss}(y_i, Y_i)$

19:                          $\nabla \theta L_i \leftarrow \dfrac{\partial L_i}{\partial \theta}$

20:                          Accumulate gradients $\nabla_\theta L_i$ for batch

21:                  **end for**

22:                  $\theta \leftarrow updateRule(\mathscr{O}, \nabla_\theta)$

23:                  $\nabla_\theta L_i \leftarrow 0$                          ▷ Reset gradients

24:          **end for**

25: **end for**

26: Validate on $\mathscr{D}val$

27: Compute evaluation metrics on $\mathscr{D}val$ **return** $\theta$

---

attention mechanism, described in Eq. (7). This mechanism refines the attention process by incorporating learned attention weights and leveraging contextual information, resulting in more informative and context-aware embeddings. Finally, the attentive pooling operation, as depicted in Eq. (8), amalgamates these refined embeddings to construct a comprehensive graph-level representation $h_{\mathscr{G}}$. Overall, GATv2's attention mechanism enhances the model's interpretability, discriminative power, and ability to capture nuanced relationships, making it a valuable tool for complex data analysis tasks like sentiment analysis. Subsequently, the updated embeddings are passed through fully connected layers (FC), ReLU activation, and softmax to obtain the predicted sentiment probabilities $y_i$. The loss $L_i$ is computed using the loss function $f^{loss}$.

Gradients are accumulated and used to update the model parameters $\theta$ using the specified update rule. After completing all epochs, the model is validated on the validation set $\mathscr{D}_{\text{val}}$, and evaluation metrics are computed.

In summary, Algorithm 2 outlines a comprehensive approach for training an inductive sentiment classification model using the GAT architecture, incorporating attention mechanisms and graph-based representations to achieve effective sentiment analysis on text graphs.

## Explainability module

In this module, classification decisions are explained based on the attention values of the learned model. It essentially consists of two steps: in the first step, a weighted graph is constructed from the learned attention values, and then adaptive filtering is applied to these attention values to obtain a subgraph pattern as explainable purposes. This module is depicted on the right bottom side of Fig. 2.

Although GNN models have various approaches for explanation, they generally rely on *ad-hoc* proxy models, which can increase complexity and lead to information loss due to indirect explanations (*Pope et al., 2019*). To address this issue, in this module, we develop two different inherent explanation models based on the learned attention values from the last layer of the previous learning module. These models provide explanations at both the edge level and the subgraph level, aiming to overcome the complexity and information loss associated with *ad-hoc* proxy model approaches like GNNExplainer (*Ying et al., 2019*).

As described in "Attention-based GNN Design for Sentiment Classification", the learning module's last layer contains learned attention values for each edge based on the model's decision. These values are used to create a weighted graph, with node values (words) included in the graph for consistency with the learning module. This graph is represented as $G_{\text{att}}(V, E)$.

This module explains the decision-making process of the model, which is based on the attention values learned in the Attentive Classification module in "Attention-based GNN Design for Sentiment Classification". This process develops an algorithm inspired by a adaptive background filtering method (*Ostu, 1979*), which is widely used in image processing problems.

This naive approach is to adaptively filter and rank the learned attention values. This solution algorithm is given in Algorithm 3.

The algorithm takes as input the trained model parameters/weights $\theta$ and the text representation graph $G(V, E)$, and it outputs the filtered explainability subgraph $G_{\text{exp}}$. The algorithm begins by computing edge attention weights $G_{\text{att}}(V, E)$ using the model parameters $\theta$ and the input graph $G(V, E)$. It then proceeds to compute a histogram and a cumulative distribution function (CDF) from these attention weights to determine an adaptive threshold for filtering edges. The threshold is chosen based on maximizing the inter-class variance between two classes defined by the CDF. Finally, the algorithm filters the edges of the input graph $G$ based on the computed threshold, resulting in the filtered explainability subgraph $G_{\text{exp}}$. This algorithm offers a systematic approach to extract

**Algorithm 3** Proposed adaptive filtering based explainability generation.

1: **Input:** $\theta$: GNN trained model parameters/weights, $G(V, E)$: Text representation graph

2: **Output:** Filtered explainability subgraph $G_{\text{exp}}$

3: Initialize $G_{\text{filtered}}$ as an empty graph

4: $G_{\text{att}}(V, E) \leftarrow$ Compute edge attention weights using $\theta$ and $G$

5: Initialize histogram $H$ with $N$ bins

6: Initialize cumulative distribution function (CDF) $C$ with $N$ bins

7: Compute histogram $H$ from edge attention weights $G_{\text{att}}$

8: Compute CDF $C$ from histogram $H$

9: Compute total sum $S$ of all edge attention weights in $G_{\text{att}}$

10: Initialize maximum inter-class variance $maxVar$ to 0

11: Initialize adaptive threshold $T$ to 0

12: **for** $t \leftarrow 1$ to $N - 1$ **do**              ▷ Iterate through possible thresholds

13:          Compute class probabilities $P_0$ and $P_1$ using CDF $C$ and threshold $t$

14:          Compute mean values $m_0$ and $m_1$ of class 0 and class 1

15:          Compute inter-class variance $var$ using $P_0$, $P_1$, $m_0$, $m_1$, and $S$

16:          **if** $var > maxVar$ **then**        ▷ Update maximum inter-class variance and threshold

17:                 $maxVar \leftarrow var$

18:                 $T \leftarrow t$

19:          **end if**

20: **end for**

21: **for** each edge $e$ in $G$ **do**

22:          **if** attention weight of $e$ in $G_{\text{att}}$ passes $T$ **then**

23:                 Add $e$ to $G_{\text{exp}}$

24:          **end if**

25: **end for**

26: **return** $G_{\text{exp}}$

informative edges from a graph, enhancing the interpretability and explainability of GNN models in text analysis tasks.

The time complexity of the proposed algorithm can be analyzed as follows. First, computing edge attention weights $G_{\text{att}}(V, E)$ takes $O(E)$ time, where $E$ is the number of edges in the input graph. Next, initializing and computing the histogram $H$ and cumulative distribution function (CDF) $C$ from the edge attention weights takes $O(E)$ time as well. The loop iterating through possible thresholds runs for $N - 1$ iterations, where $N$ is the number of bins in the histogram. Inside this loop, computing class probabilities, mean values, and inter-class variance for each threshold takes $O(1)$ time per iteration. Therefore, the total time complexity of the algorithm is $O(E + N)$, where $E$ is the number of edges and $N$ is the number of bins in the histogram.

As for space complexity, the algorithm requires space to store the input graph $G(V, E)$, the model parameters/weights $\theta$, the edge attention weights $G_{att}(V, E)$, the histogram $H$ and cumulative distribution function (CDF) $C$, and variables for computations such as mean values, class probabilities, and thresholds. The space complexity is dominated by the input graph and the edge attention weights, resulting in a space complexity of $O(E)$ for the input graph and $O(E)$ for the edge attention weights, leading to a total space complexity of $O(E)$ for the algorithm.

## EXPERIMENTAL STUDIES

This section presents experimental studies to address the following research questions:

- RQ1: What is the capacity of a pre-trained language model to generate embedding vectors at a layer-by-layer level, and which layers exhibit higher levels of semantic discrimination?
- RQ2: What is the dependency coverage rate for graph representation in Turkish texts using attention weights of a pre-trained language model, specifically targeting the challenges highlighted in "Handling of suffixes"?
- RQ3: How does the representation model derived from the language model affect classification performance, and what are the performance disparities between attention-based and non-attention-based GNN models?
- RQ4: To what extent is the most successful model explainable, considering both qualitative and quantitative aspects of its explainabilities?

We conducted the following experiments using Turkish sentiment classification benchmark datasets, respectively:

- **(i) Layer-wise embedding evaluation:** This experiment involved visualizing the embedding vectors generated at a layer-by-layer level by a pre-trained Turkish language model. It aimed to investigate the capacity of the model to capture semantic information across different layers, in line with RQ1.
- **(ii) Dependency graph accuracy:** We measured the accuracy of dependency relationships using labeled data, as mentioned in RQ2. The goal was to assess how well the model can identify and represent dependencies within the text.
- **(iii) Classification performance comparison:** This experiment focused on comparing the performance of common graph-based learning models using graph representations derived from Turkish sentiment benchmark datasets. It aimed to address RQ3 by evaluating the effectiveness of attention mechanisms in improving classification performance.
- **(iv) Evaluation of explainability:** In response to RQ4, we evaluate the explainability of the model using two common metrics as known as fidelity and sparsity. The objective is to assess how well the model's predictions could be interpreted and explained in both qualitative and quantitative terms.

These experiments were carefully designed to provide a detailed understanding of the limitations and capabilities of our framework using performance metrics, and provide valuable insights into the nuances of language modelling and graph representation in Turkish sentiment analysis.

### Benchmark datasets

The experiments have been performed on three real-world Turkish sentiment datasets with varying sizes. These are briefly described as follows:

- **TripAdvisor:** The dataset comprises 42,000 hotel reviews from the TripAdvisor web page (*Büyükeke, Sökmen & Gencer, 2020*), with sentiment labels in three categories. However, it is an unbalanced dataset, with 80% of the reviews being positive. To address this issue, we randomly selected 1,250 samples with an equal number of reviews in each class for this study.
- **ImdbFilmReview:** Film reviews from IMDB (*Amasyalı et al., 2012*), which contain three traditional sentiment categories: positive, negative and neutral.
- **BlogPosts:** Users' sentiments from blog pages (*Amasyalı et al., 2012*).

The basic statistics of these datasets are listed in Table 1. Please note that the volume of *TripAdvisor* dataset is much larger than the others. For this reason, this dataset was taken as a reference in some outputs given for demonstration purposes.

Furthermore, it is important to note that the BlogPosts dataset contains a higher number of classes due to its mood-based labeling approach, which differs from the traditional positive, negative, and neutral labels found in datasets like IMDB and TripAdvisor. The inclusion of this dataset is intended to evaluate the adaptability and robustness of the proposed solution in scenarios involving more nuanced label distributions. Specifically, the goal is to observe the performance of the GAT architecture, used as the backbone, compared to other architectures under default configurations across diverse class distributions. This allows for a comprehensive assessment of the model's ability to handle complex or unconventional classification tasks, such as mood-based labeling.

### Comparison models

As stated in "Graph Construction Module", the framework proposes solutions for constructing graphs from Turkish texts, constructing an attentive model from these graphs, and explaining predictions. The main task is to build a successful model based on the graph representation. To evaluate the performance of the model, the two popular no-attention based GNN models were used:

- **GCN:** In GCNs (*Kipf & Welling, 2016*), a specialized model architecture is employed for tasks involving graph data, such as node classification and graph-level prediction. Unlike traditional neural networks, GCNs operate directly on graph structures, leveraging node features and their relationships. This approach allows GCNs to capture complex

**Table 1 Dataset statistics.**

| Dataset | Num of classes | Train size | Test size |
|---|---|---|---|
| TripAdvisor[1] | 3 | 1,000 | 250 |
| ImdbFilmReview[2] | 3 | 100 | 20 |
| BlogPosts[3] | 4 | 35 | 5 |

Notes:
[1] https://github.com/ahmeteke/turkish-tourist-reviews-data-.
[2] https://doi.org/10.5281/zenodo.14823173.
[3] https://doi.org/10.5281/zenodo.14823096.

dependencies and patterns within graphs, making them effective for tasks like social network analysis, molecular structure prediction, and recommendation systems.

- **GIN:** Unlike GCNs that operate on fixed graph structures, GINs (*Xu et al., 2018*) dynamically consider different graph isomorphisms, enabling a more nuanced analysis of graph properties. This dynamic approach allows GINs to excel in tasks requiring precise graph matching and classification, such as molecular structure comparison, subgraph detection, and graph similarity assessment.

Please note that all of these models aim to learn graph representations and do not use the attention mechanism.

## Evaluation metrics

For attentive classification module, macro F1-score (*Hand & Christen, 2018*) is preferred as evaluation metric due to several reasons. First, it addresses the challenge of class imbalance commonly observed in real-world graphs. Since some classes may have significantly more instances than others, the macro F1-score treats each class equally, preventing dominance by the majority class. Second, GNNs operate on local structural information within graphs, but global graph structure also matters. By encouraging a balanced consideration of all classes, macro F1-score promotes learning from both local and global contexts, improving model generalizability.

Technically, the F1-score combines precision and recall, providing a balanced measure for binary classification. For multi-class problems, we use variants like micro F1-score and macro F1-score. Here's how they are defined:

Precision measures how many of the "positive" predictions made by the model were correct:

$$Precision = \frac{True\ Positives\ (TP)}{TP + False\ Positive\ (FP)}. \tag{9}$$

Recall measures how many of the positive class samples present in the dataset were correctly identified by the model:

$$Recall = \frac{TP}{TP + False\ Negatives\ (FN)}. \tag{10}$$

The F1-score is the harmonic mean of precision and recall:

$$F1 = 2 \cdot \frac{Precision \cdot Recall}{Precision + Recall}. \tag{11}$$

For multi-class problems, we compute the macro-averaged F1 score by taking a simple average of the class-wise F1 scores:

$$MacroF1 - score = \frac{1}{n} \sum_{i=1}^{n} F1_i. \tag{12}$$

where $n$ is the number of classes.

On the other hand, to empirically evaluate Explainability module, two commonly used metric is used. These are the fidelity and sparsity (*Pope et al., 2019*) metrics.

Fidelity and sparsity are two common metrics for evaluating the explainability of GNN. Fidelity measures how well the explanation preserves the prediction of the original model, while sparsity measures how concise the explanation is.

Formally, let $G = (V, E)$ be the original graph, and $G' = (V', E')$ be the subgraph generated by the explanation method. Let $f(G)$ be the prediction of the GNN model on $G$, and $f(G')$ be the prediction of the same model on $G'$.

Here, two different fidelity calculation approaches (*Amara et al., 2022*) are referred as follows and their harmonic mean is calculated as the fidelity value as shown in Eq. (15), where $w_+$ and $w_-$ are the weights for $Fid_+$ and $Fid_-$, respectively.

$$Fid_+ = f(G) - f(G - G') \tag{13}$$
$$Fid_- = f(G) - f(G') \tag{14}$$
$$Fidelity = \frac{(w_+ + w_-)}{\left( \frac{w_+}{Fid_+} + \frac{w_-}{1-Fid_-} \right)}. \tag{15}$$

On the other hand, the calculation of the sparsity values of the generated explanations is calculated as in Eq. (16)

$$Sparsity = \frac{|G.V| - |G'.V|}{|G.V|}. \tag{16}$$

This metric also ranges from 0 to 1, where a higher value means a higher sparsity.

## Implementation details

Our experimentation was conducted using the Python programming language, primarily selected for its open-source nature, robust community support, and extensive resource availability. Specifically, we leveraged the *transformer* library for experiments involving pre-trained language models and employed the *pytorch-geometric* library for experiments concerning GNN models. Additionally, authors developed their custom Python code to implement technical contributions such as Turkish text suffix normalization, graph construction, and adaptive attention filtering, as detailed in "Problem Statement".

For our runtime environment, we utilized the Google Colab+ machine, which offers a Tesla K80 GPU, 12 GB of RAM, and an Intel Xeon CPU clocked at 2.20 GHz.

In our study, we utilized *BERTurk* as a pre-trained language model specifically tailored for Turkish text. This choice was motivated by its status as the most comprehensive transformer-based language model trained on Turkish data, providing a robust foundation for our experiments.

## Results

### Layer-wise embedding evaluation

As stated in "Graph Construction Module", the graph construction process mainly rely on a pre-trained language model. To evaluate the quality of the vectors generated by the language model, as a response to RQ1, we conducted a visualization-based experiment on TripAdvisor dataset, the largest benchmark dataset. This experiment aims to demonstrate how the pre-trained language model embeddings capture semantic relationships without additional fine-tuning. Rather than training the language model on the dataset, this approach evaluates its effectiveness in inference mechanism, as described in "Context-Aware Word Embeddings", showcasing its capability to capture semantic discrimination in sentiment analysis.

For the sake of simplicity, the document vectors are averaged naively over the word vectors and this process is done separately for each layer to obtain document embedding vectors.

Since the embedding vectors in the 12 layers of the model are 768-dimensional, we use T-SNE (*Van der Maaten & Hinton, 2008*) method to visualize them in two-dimensional space. Figure 3 depicts how the different sentiment classes are positioned in space for each layer. It is clearly observed that the documents are grouped more accurately as we move towards the higher layers. This confirms that the embedding vectors from the final layers provide the highest levels of semantic discrimination. These findings align with observations from previous studies conducted in other languages (*Htut et al., 2019*; *Limisiewicz, Rosa & Mareček, 2020*), validating that semantic discrimination in higher layers is also applicable to Turkish sentiment data, as outlined in RQ1.

### Dependency graph accuracy

Unlike embedding vectors, attention matrices are located in each layer and in the heads of these layers. In order to evaluate these attention matrices, an accuracy metric is performed for each layer/head selection based on a labeled Dependency Graph dataset in the literature (*Marşan et al., 2022*). Experimental results are shown in Fig. 4. When we look at these results in general, it is observed that the accuracy values in the early layers are higher, that is, the syntactic relations are better modeled. This is in parallel with the experimental results of studies on other languages in the literature (*Htut et al., 2019*; *Limisiewicz, Rosa & Mareček, 2020*). In other words, this is the first time that the results of this experiment have been shown in the Turkish language. It is also observed that the selection of a single

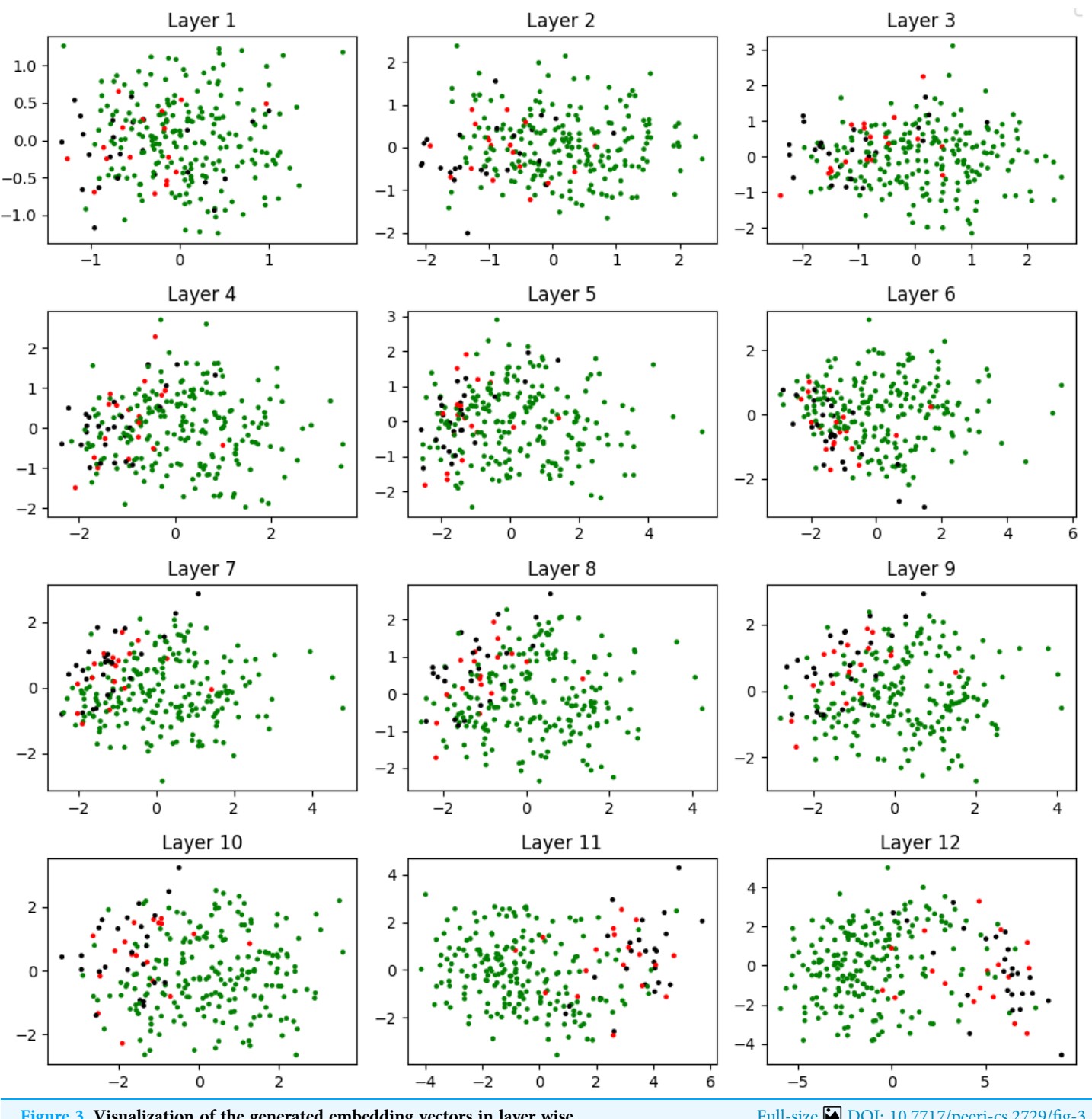

**Figure 3 Visualization of the generated embedding vectors in layer wise.**

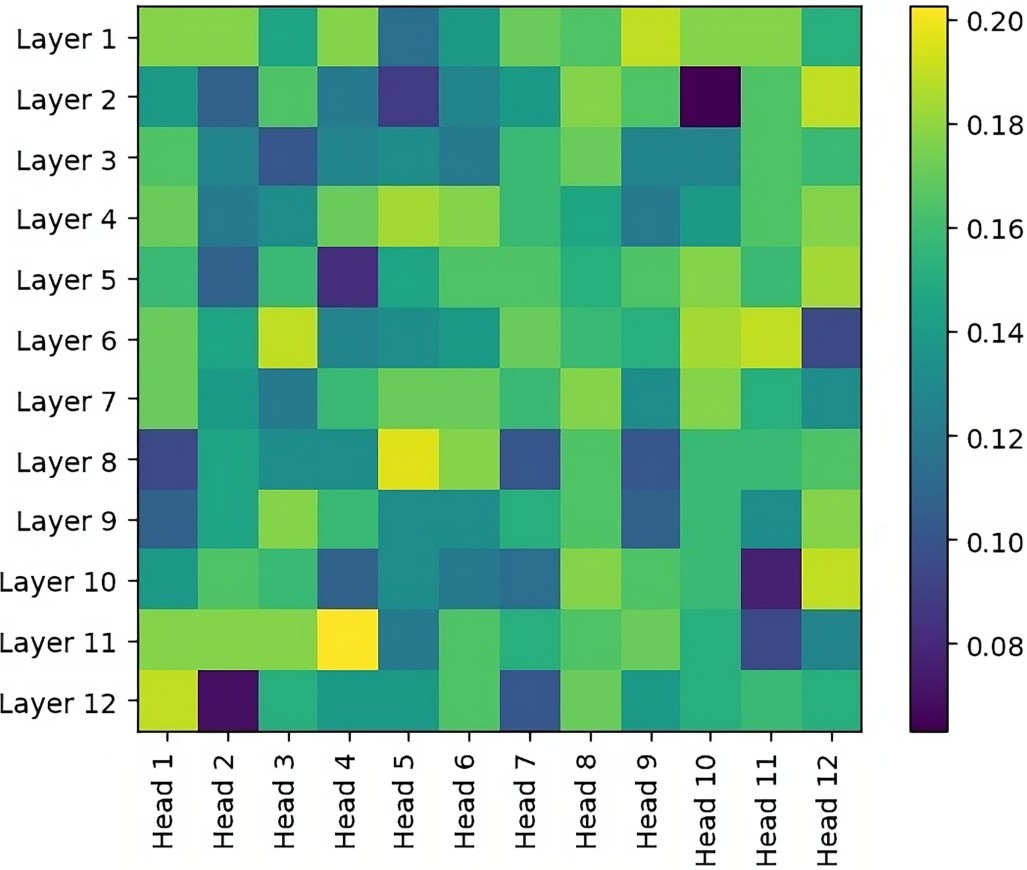

**Figure 4 Dependency graph accuracy wrt varying head/layer in the pre-trained BERT model.**

layer/head achieves max 0.2 success. This situation shows that more than one setting selection should be ensemble.

Please note that the search space for the optimal layer/head combination grows exponentially with the number of layers and heads as $2^{\#layers*\#heads}$. To avoid increasing computational complexity, we conduct experiments using overall average attention values.

### Classification performance comparison

As explained in "Attentive Classification Module", the *Classification* module mainly relies on an attention based GNN model. To determine the most successful GNN model, a set of experiments are conducted over three real-world Turkish sentiment datasets. Additionally, a comparative analysis is carried out, considering the size of the data sets. The data sets are divided into those that use attention-based techniques and those that do not, in order to observe any differences in performance.

As detailed in "Proposed Approach", *TurkSentGraphExp*, which integrates a graph-based learning mechanism to model inter-word relationships in sentiment analysis, underscores the importance of this structure for explainable sentiment analysis in Turkish

texts. In this context, four existing graph-based models (GCN, GCI, GAT, GATv2) capable of learning from graph input representations were compared in terms of sentiment recognition, as shown in Table 2. Notably, the GAT variants (*Veličković et al., 2017*) GATv2 (*Brody, Alon & Yahav, 2021*), utilized as the backbone of *TurkSentGraphExp*, outperformed the other models with margins of 0.1, 0.8, and 0.53, respectively. These results highlight the effectiveness of attention-based GRL models in recognizing sentiment.

Additionally, it is observed that even the most successful model on *ImdbFilmReview* and *BlogPosts* data, where the number of documents is low, remains at 40–50% levels. However, when the size of the dataset in *TripAdvisor* data is large, it is observed that the success of the model is quite high. This observation indicates that the size of the training set may affect the performance of the model.

Finally, in order to ascertain whether the learning process of the model is exhibiting signs of overfitting, the loss values are presented in Fig. 5. Upon examination of these values, it becomes evident that there is no discernible difference in loss between the training and validation data. This observation indicates that the learning model is not exhibiting a tendency towards overfitting.

### Evaluation of explainability

To answer RQ4, the explainability of the proposed model is evaluated in qualitatively and quantitatively. For the qualitative evaluation, observations are made through some specific extreme case studies, while for the quantitative evaluation the metrics described in "Evaluation Metrics" are used. Specifically, for the qualitative evaluation, we focused on situations with different emotional expressions such as neutral or mixed sentiment reviews. This is because traditional approaches modelled without the use of graphs/dependencies fail to classify texts in this form (*Demirci, Keskin & Doğan, 2019*; *Yıldırım et al., 2015*). For example, let's consider the following comments containing three neutral or mixed emotions for demo purposes:

- Otel (The hotel) konforlu (comfortable), ancak (but) hizmet (service) yetersiz (insufficient).
- Oda (The room) geniş (spacious), fakat (yet) temizlik (cleanliness) sorunlu (problematic).
- Personel (Staff) yardımsever (helpful), lakin (however) yemekler (meals) lezzetsiz (tasteless).

The explanations of these three interpretations derived from the sentiment classification predictions are shown in Figs. 6–8. It is observed that the sentiment phrases are detected with high weights by the proposed framework, while the overall discriminability of the explanations produced by the GNNExplainer model is low.

From a quantitative perspective, two common approaches are used in the literature to measure the explanation quality of models: fidelity and sparsity. By applying these approaches as formulated in "Evaluation Metrics", the explanation quality of the proposed

**Table 2 Model comparison with macro-F1 scores on benchmark datasets.**

| Attn | Model | TrpAdvisor | Imdb | BlgPost |
|------|-------|-----------|------|---------|
| No | GCN | 0.11 | 0.50 | 0.30 |
| No | GIN | 0.21 | 0.16 | 0.31 |
| Yes | Ours/w*GAT* | **0.74** | 0.48 | 0.32 |
| Yes | Ours/w*GATv2* | 0.70 | **0.58** | **0.32** |

**Note:**
Values in bold denote that the GAT variants outperformed the other models with margins of 0.1, 0.8, and 0.53, respectively.

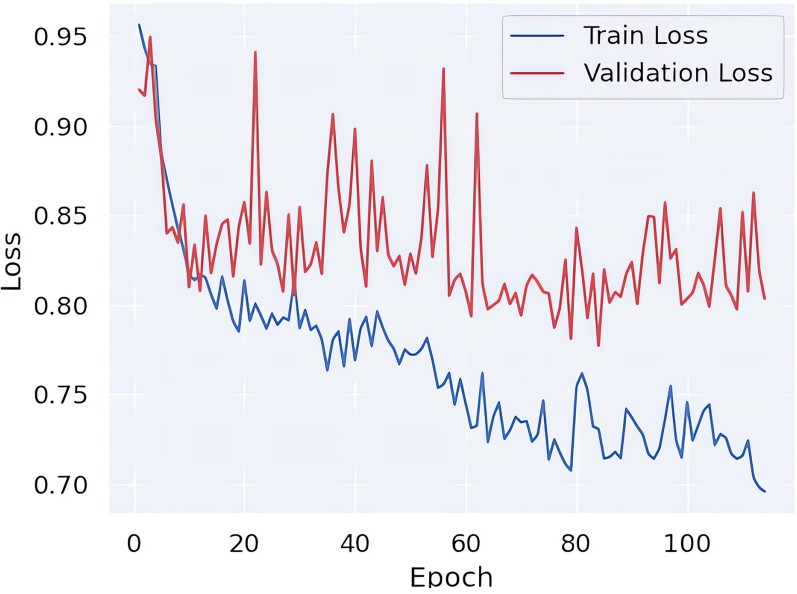

**Figure 5 Train *vs* validation loss curves.**

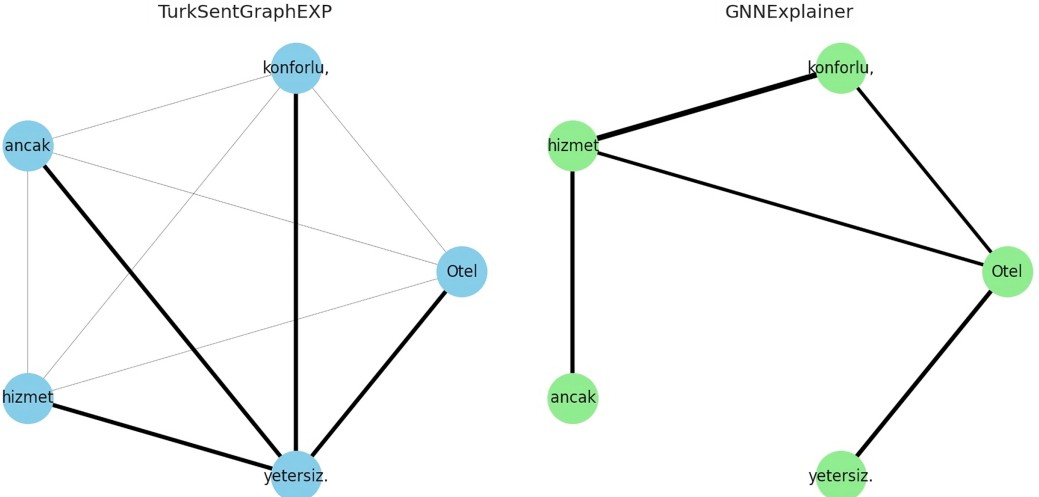

**Figure 6 Explaining sentiment classification of "Otel (The hotel) konforlu (comfortable), ancak (but) hizmet (service)".**

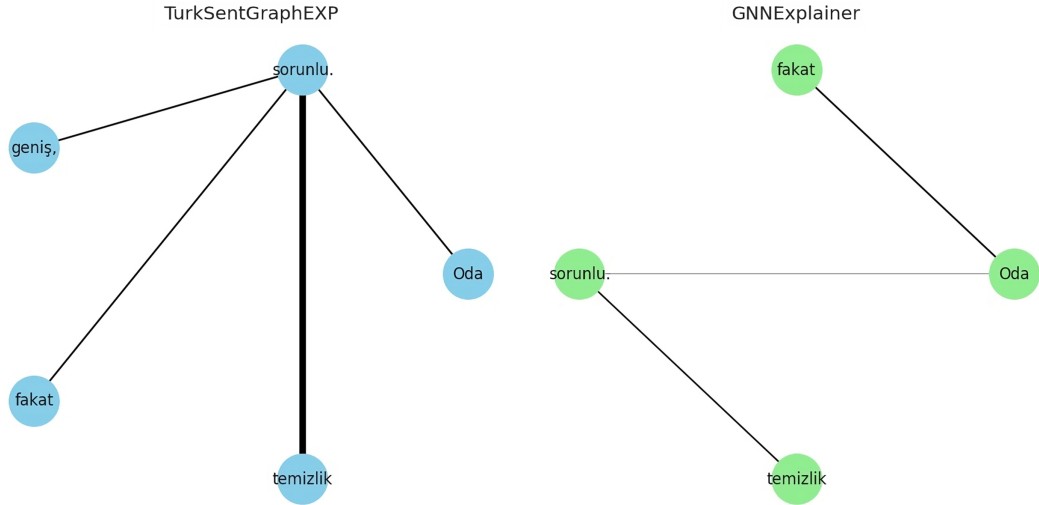

**Figure 7 Explaining sentiment classification of "Oda (The room) geniş (spacious), fakat (yet) temizlik (cleanliness) sorunlu (problematic)".**

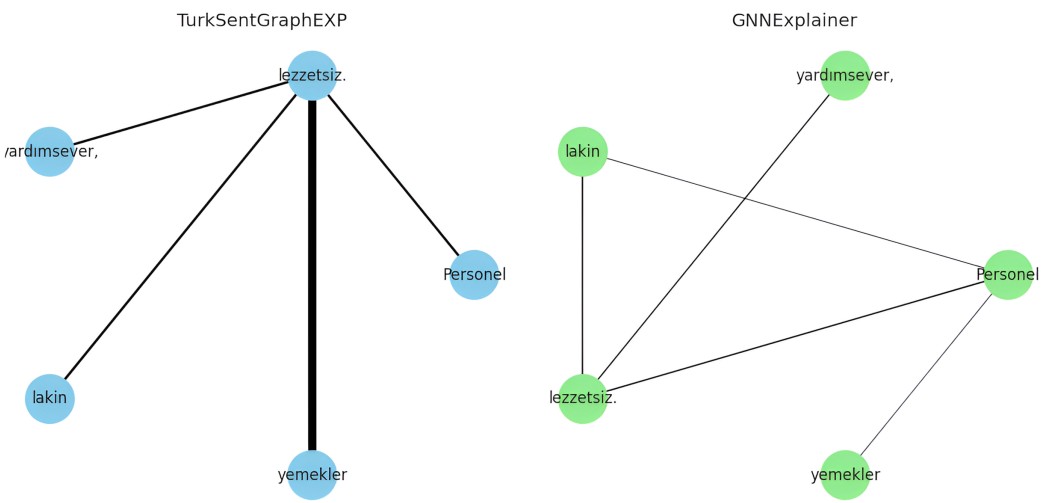

**Figure 8 Explaining sentiment classification of "Personel (Staff) yardımsever (helpful), lakin (however) yemekler (meals) lezzetsiz (tasteless)".**

model is observed in Fig. 9. With this observation, the results are compared with a popular GNN explainability model called GNNExplainer. According to these results, the higher fidelity value of the proposed framework according to the changing sparsity rate indicates that more faithful explanations are produced in terms of the quantity of explanations. On the other hand, since the proposed model works inherently, there is no need for an *ad-hoc* model, it is obvious that the model produces faster results with less complexity. Such quantitative observations show that the proposed model is effective.

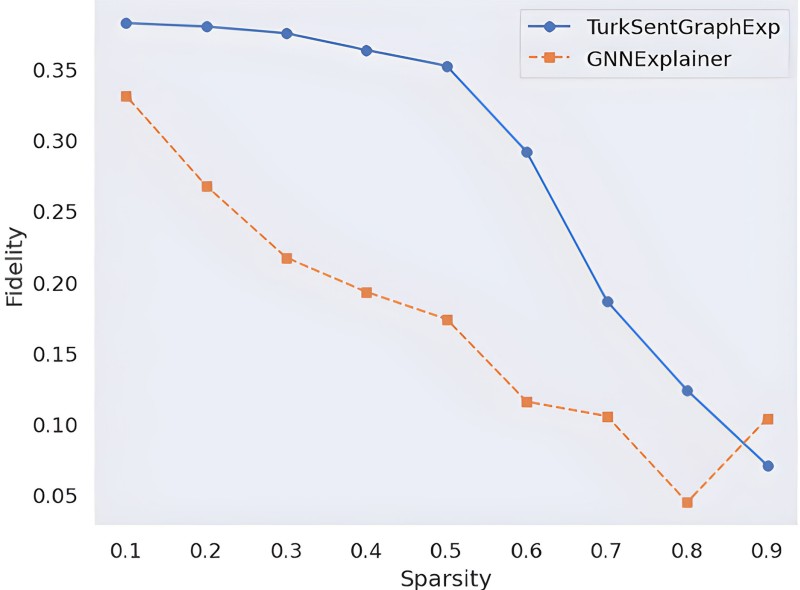

**Figure 9 Quantitative explainability score comparison.**

## CONCLUSION

In this study, we propose an inherent graph-aware explainability framework for Turkish Sentiment Analysis tasks, named *TurkSentGraphExp*, which applies attention-based GNN models to explain predictions inherently in graph-aware.

Particularly from a technical point of view, we first constructed graph representations of the documents by utilizing a pre-trained language model in order to handle out of vocabulary tokens that may occur due to the fact that the Turkish language is agglutinative. Then, by modelling these representations with inductive setting, we built a GRL model using an attention-based GNN model. Adaptive filtering over the inherent explainable attention scores of the attention-based model is implemented by making an analogy with the background filtering used in image processing applications.

For the first time in Turkish NLP domain, the experimental results show that the attention-based models used in the proposed framework are not only inherently explainable in Turkish sentiment datasets but also successful in terms of model performance on three real-world sentiment datasets.

In our future work, we will analyze the outputs of the state-of-art models utilizing explainability approaches and aim to build more accurate models by using prompt engineering methods. Moreover, for the explainability side, we plan to make some integrations in order to provide a high-level explainability at the phrase level.

### Funding

The authors received no funding for this work.

## Competing Interests

The authors declare that they have no competing interests.

## Author Contributions

- Yasir Kilic conceived and designed the experiments, performed the experiments, analyzed the data, performed the computation work, prepared figures and/or tables, authored or reviewed drafts of the article, and approved the final draft.
- Cagatay Neftali Tulu conceived and designed the experiments, analyzed the data, prepared figures and/or tables, authored or reviewed drafts of the article, and approved the final draft.

## Data Availability

The third party datasets used are available at:

- TripAdvisor: https://github.com/ahmeteke/turkish-tourist-reviews-data-.

- ImdbFilmReview: Amasyali, F., & Kaya, A. A. (2025). Turkish Movie Reviews Dataset. Zenodo. https://doi.org/10.5281/zenodo.14823173.

- BlogPosts: Amasyali, F. (2025). Bloggers Mood Dataset in Turkish. In EMO Bilimsel Dergi (Vol. 2, Number 4, pp. 95–104). Zenodo. https://doi.org/10.5281/zenodo.14823096.

The source code of the project is available at Zenodo: KILIÇ, Y., & Tulu, C. N. (2025). TurkSentGraphExp: An Inherent Graph Aware Explainability Framework From Pre-trained LLM For Turkish Sentiment Analysis. Zenodo. https://doi.org/10.5281/zenodo.14760110.

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
