# Peer review of "TurkSentGraphExp: an inherent graph aware explainability framework from pre-trained LLM for Turkish sentiment analysis"

_PeerJ Computer Science, doi:10.7717/peerj-cs.2729_

## Round 0.1 · original submission · Major Revisions

Dear Authors,

Thank you for the submission. The reviewers’ comments are now available. It is not suggested that your article be published in its current format. We do, however, advise you to revise the paper in light of the reviewers’ comments and concerns before resubmitting it. The followings should also be addressed:

1. Please pay special attention to the usage of abbreviations. Spell out the full term at its first mention, indicate its abbreviation in parenthesis and use the abbreviation from then on.
2. Equations should be used with correct equation number. Please do not use “as follows”, “given as”, etc. Explanation of the equations should also be checked. All variables should be written in italic as in the equations. Their definitions and boundaries should be defined. Provide proper reference to the governing equations.
3. Equations are part of the related sentences. Attention is needed for correct sentence formation.
4. Please recheck all the definition of variables used in the equations and further clarify these equations. All variables should be written in italic as in the equations.
5. Minor grammar and writing style errors should be corrected. Please pay special attention for correct writing, adjusting, and formatting. Especially pay special attention for the usage of space character.

Best wishes,

Reviewer 1 ·

Basic reporting

Abstract:
• The abstract did not mention that the proposed solution is based on graph representation learning. It is suggested that the graph representation learning is included in the abstract.
• The abstract should state some key results.
• The abstract should state the three types of sentiment: positive, negative, and neutral.

Introduction
• The introduction is too lengthy. Re-arrange some of the content as research background.
• The research problem is not clear. Is the purpose of the new solution to reduce dependency on the database, to tackle the unique nature of the complex agglutinative structure of the Turkish language, or to improve the sentiment analysis performance?
Literature review
• There is a lack of explanation for the issue of complex Agglutination of the Turkish language on emotion recognition in text.
• At line 251: “In the study, both instance-level and model-level graph-based ExAI methods were tested on various datasets, specifically for sentiment analysis tasks”. However, the rest of the paper did not discuss instance-level and model-level graphs. If this line refers to the literature, there was no citation.

Problem Description
• Line 270: “Our goal is to predict the sentiment labels that are related to the review and to give corresponding explanations in phrase level.” This goal is not reflective of the title and the abstract. For example, the dealing with the complex agglutinative structure of the Turkish language.
• Line 312: This paragraph needs citations to verify the use of the BERTurk model.
• Line 317: This paragraph needs citation to verify using the BERTurk model.
• Line 374: it was stated by authors that negation suffixes play a crucial role in sentiment analysis. Yet, in an earlier paragraph, it was stated that the model from BERTurk is modified to only focus on the stem and ignore suffixes. It seems that the authors are giving contradictory statements on the role and importance of suffixes
• Line 378: authors select the Dependency Graph for their proposed solution. However, the dependency graph was not discussed in introduction and LR section. It is unclear from where the authors decided to use the Dependency Graph as the best solution in their research.

Experimental design

Experimental studies
Line 570: Unlike the traditional positive, negative, and neutral labels found in the IMDB and TripAdvisor datasets, BlogPosts is labeled according to mood. Is the use of BlogPosts not suitable as the model developed using BERTurk is based on positive, negative, and neutral? As such BlogPosts is not suitable for training and testing.
Table 1 states that the selected database input will be used to perform training and some for testing. If that were the case, what is the purpose of having a pre-trained model using BERTurk? Are the authors performing some form of model adaptation?
Line 631: The authors claimed to have conducted a visualization-based experiment on the TripAdvisor dataset, the largest benchmark dataset. However, the result in Figure 2 is based on a Pre-trained BERTurk model. They should show the result of the Pre-trained TripAdvisor dataset instead of the Pre-trained BERTurk model.
At the end of section 4.5.1., the authors did not fully answer RQ1: What is the capacity of a pre-trained language model to generate embedding vectors at a layer-by-layer level, and which layers exhibit higher levels of semantic discrimination? For example, they did not state which layers exhibit higher levels of semantic discrimination.
Line 646: “This is in parallel with the experimental results of studies on other languages in the literature ()” has missing citations.
Line 666: “This observation indicates that the size of the training set may affect the performance of the model”. Such a statement raises the question of why the authors did not use the same data size for all real-world examples. Maybe the authors can use other sources like social media postings such as Facebook or X as the source of data? Having equal data size helps to validate the effectiveness of the proposed model.

Validity of the findings

What is TurkSentGraphExp? It was stated in the title but mentioned only once in the abstract, once in Figure 6, and once in the conclusion. The authors need to explain the working model of TurkSentGraphExp.
The article did not provide findings as to the effectiveness of the proposed approach for dealing with the complex agglutinative structure of the Turkish language. For example, how does the proposed model fare better than existing models like BERTurk when recognizing sentiment? Or how does the proposed model better classify the effect of suffixes on the meaning of the text?

Reviewer 2 ·

Basic reporting

General:

The paper is well-structured and easy to read, with clear. However, there are some repetitions in the use of abbreviations and their full forms. For instance, 'Graph Attention Networks (GATs)' is repeated on lines 256 and 288–289. Since the abbreviation has already been defined, it would be better to use 'GATs' in subsequent mentions.
In line 436, there is a minor typo: a whitespace is needed between ',' and 'describe.'
The paper provides good literature references and sufficient field background/context. Figures and tables are well-prepared, and raw data is shared. It also provides clear definitions, including formal mathematical representations.

Literature Background and Context:

The article includes hypotheses relevant to the field, one of which proposes constructing a graph on text reviews to enhance explainability and support classification decisions. These are followed by well-defined research questions and appropriate experiments. Results are supported with relevant figures and interpretation.

Article Structure:

The title and conclusion introduce TurkSentGraphExp, but the proposed model is not clearly named in the manuscript, particularly in the approach section. Additionally, the paper lacks a distinct "Problem Statement" and "Approach" section. Dividing the problem description into these sections would make the tasks and methods clearer.

Self-Contained Results:
The results are self-contained and support the hypotheses. However, to answer RQ3 comprehensively, it would be better to compare the proposed model's classification performance in sentiment analysis with state-of-the-art (SOTA) models mentioned in the literature review.

Experimental design

Originality and Scope:

The manuscript aligns well with the scope of the journal, proposing a novel model for sentiment analysis using explainable AI. The approach combines a pre-trained model derived from BERT with Graph Neural Networks (GNN), addressing challenges related to the reliance on black-box models.

Research Questions (RQs):

The research questions are well-defined, relevant, and meaningful. The authors aim to make the model more transparent by leveraging explainable AI techniques for generating results. The study attempts to fill a significant knowledge gap in the field.

Methodological:

The results presented in the manuscript adequately address the research questions. However, to strengthen the investigation, the authors should:
- Compare the model’s performance with state-of-the-art (SOTA) methods to demonstrate its efficacy.
- Conduct statistical tests to determine whether the proposed model provides significant improvements in classification performance.

Clarity of Methods:

The model's approach is explained clearly within the problem description. However, dividing this section into a "Problem Statement" and an "Approach" would improve clarity and make the methodology more understandable for readers.

Validity of the findings

Impact and Novelty:

The authors claim that the proposed model is the first in the field of sentiment analysis to provide explainable AI capabilities for the Turkish language. This is an innovative and impactful approach that leverages a language model to construct a graph for explainability, instead of relying on traditional tokenization methods. The rationale and benefits to the literature are clearly stated.

Robustness and Statistical Soundness:
The results presented in the manuscript are relevant and provide answers to the research questions. However, the authors need to perform statistical tests to determine whether the observed results are significant when compared to other models. This will strengthen the validity of the findings in terms of model performance.

Conclusions and Rationale:

The conclusions are well stated, address the research questions clearly, and align with the supporting results. However, while the conclusion mentions the name of the model, this name is not explicitly mentioned in other sections of the manuscript. Including the model name in earlier sections, particularly in the methodology, would improve consistency and clarity.

---

## Round 0.2 · accepted · Accept

Dear Authors,

One of the previous reviewers, who had requested a minor revision, did not respond to the invitation to review the revised paper in time. However, the other reviewer considered that the author had addressed the majority of the comments from the previous review, and that the paper had been sufficiently improved. Your revised paper was therefore accepted for publication, subject to very minor edits provided by Reviewer 1.

Best wishes,

Reviewer 1 ·

Basic reporting

No comment.

Experimental design

No comment.

Validity of the findings

The authors have addressed most of the comments from the previous review. However, here are some concerns.

At line 601: To evaluate the performance of the model, the two popular no-attention based GNN models were used.
Comment: Prior to this, the authors did not mention or discuss the use of GNN model for evaluation purposes. In which existing work stated that these two are popular GNN models? Add the citation.

At line 699: Notably, the GAT variants (56)GATv2 (8)), utilized as the backbone of TurkSentGraphExp, outperformed the other models with margins of 0.1, 0.8, and 0.53, respectively.
Comment: How was the margin derived? The result for TripAdvisor shows GAT as the highest and not GAT v2. Please clarify and correct this.